# Teaching Practical Skills in Anesthesia, Intensive Care, Emergency and Pain Medicine—What Is Really Relevant for Medical Students? Results of a German National Survey of Nearly 3000 Anesthesiologists

**DOI:** 10.3390/healthcare10112260

**Published:** 2022-11-11

**Authors:** Franziska Busch, Andreas Weissenbacher, Sebastian N. Stehr, Tobias Piegeler, Gunther Hempel

**Affiliations:** Department of Anesthesiology and Intensive Care, University of Leipzig Medical Center, Liebigstrasse 20, 04103 Leipzig, Germany

**Keywords:** learning objectives, medical education, medical studies, online survey, anesthesiology and intensive care, practical Skills, curriculum development, assessment & education

## Abstract

As a part of a major reform of the medical curriculum in Germany, the national catalogue of learning objectives is being revised with the focus shifting from theory-based learning to teaching practical skills. Therefore, we conducted an online survey to answer the question, which practical skills are essential in anesthesia. Participants were asked to rate the relevance of several skills, that medical students should be able to perform at the time of graduation. A total of 2898 questionnaires could be evaluated. The highest ratings were made for “bringing a patient into lateral recumbent position” and “diagnosing a cardiac arrest”. All learning objectives regarding regional anesthesia were rated as irrelevant. Furthermore, learning objectives like “performing a bronchoscopy” or “performing a rapid sequence induction” had low ratings. In the subgroup analysis, physicians with advanced training and those who were working at university hospitals rated most skills with higher relevance compared to others. Our survey provides a good prioritization of practical skills for the development of new curricula and assessment frameworks. The results can also help to establish our discipline as a cross-sectional subject in competency-based medical education, thus further increasing the attractiveness for medical students.

## 1. Introduction

In the past decades, medical school curricula have progressed from mostly fact- and theory-based to a more practical, skill-based approach. As a result, competency-based medical education has become increasingly important [1,2,3,4]. In some parts, the necessity for considerable structural changes in the field of medical education has become evident [5,6]. The German Council of Science and Humanities first recommended this new focus on competency-based education in 2014. The implementation of these changes has subsequently been called for by the Federal Ministry of Education and Research with the publication of the master plan for the reform of medical studies in 2017 [7,8].

The suggested changes included an adjustment of the National Competency Based Catalogues of Learning Objectives for Undergraduate Medical Education (NKLM in German), which defines future core competencies and activities medical students need to achieve in order to be eligible for medical licensing in Germany [9,10]. Already in the first edition, a comprehensive approach was chosen without assigning learning objectives to specific disciplines. The second version of the NKLM—supported by the German Society of Anesthesiology and Intensive Care Medicine (DGAI in German)—was recently published [11]. The planned further development of the NKLM defines a core curriculum (80%) for medical studies which will be the same at all medical faculties in Germany. The remaining 20% of the curriculum can be filled up by the different faculties with individual priorities. However, the actually desired (and necessary) skills and competencies as well as the required level of knowledge remain vague, even in this official government publication. Nevertheless, due to the legal requirements, almost all medical faculties in Germany are required to revise their curricula or even develop them from scratch in the next few years.

The majority of undergraduate anesthesiologic training in Germany takes place in form of lectures, seminars and practical training. However, the respective weighting of the teaching formats and the number of hours varies between the different universities [12]. This also applies to teaching and assessing practical skills. Today, all students in Germany must provide a “certificate of achievement” in the fields of anesthesiology, emergency medicine, palliative care and pain medicine as part of their medical studies in order to be able to register for the final state examinations. Especially regarding the increasing orientation towards competencies, practical training and practical examinations have gained in importance at many universities in recent years. It is important to understand that performing or teaching a practical skill alone is only a small element of a true competency in the corresponding medical field [13]. Competence as such is much more comprehensive and complex and includes various dimensions. In addition to pure technical skills (physical examination skills, procedural skills), it also includes cognitive aspects, the context, affective/moral and integrative aspects, as well as several more [13]. This must always be considered in teaching and in the development of new curricula. Competency-based medical education in turn does not define specific learning strategies or formats, but provides a clear description of intended outcomes [14].

Without question, a key aspect of anesthesia practice is the ability to perform practical procedures efficiently and safely [15]. Therefore, as a first step the aim of this study was to generate data on the practical skills German anesthesiologists consider of importance for medical students during their formal education. The results can be used to develop a future national curriculum. Additionally, they also provide insight into components that are considered important in the teaching of competencies. For this purpose, an online survey among anesthesiologists in Germany assessing the rating of practical skills as being relevant or not in the fields of anesthesia, intensive care, emergency or pain medicine, was conducted. The data was used to create an overview of skills that every student should have mastered by the end of medical school and which, at the same time, every graduate can apply when starting postgraduate training. In addition, the respondents’ age, qualification and workplace were recorded in order to evaluate their statistical impact. Education of medical students should obviously be suitable for all forms of later employment and not only for university hospitals where these curricula are usually developed. Overall, the data of the current study are meant to be utilized in teaching and curriculum development in the future.

## 2. Materials and Methods

### 2.1. Selecting the Learning Objectives

As a first step, practical skills considered to be relevant for an anesthesiologist, were defined. For this purpose, the NKLM and the DGAI catalogue for learning objectives for medical students were screened [16]. The NKLM is based on legal requirements of the licensing regulations for physicians and is continuously revised in the context of the current reform of medical studies. The fall 2019 version of the NKLM was used. In addition, curricula for final year medical students at various German Medical Schools and current literature regarding this matter were screened. Competencies and practical skills were arranged according to content and were formulated in a uniform manner.

### 2.2. Online Questionnaire

An online questionnaire for data collection was created using the SoSci Survey software (SoSci Survey GmbH; Munich, Germany) and made available to participants via an internet link [17]. The study was submitted to the data protection officer and the Institutional Review Board of the Medical Faculty of the University of Leipzig (Liebigstrasse 18, 04103 Leipzig-Germany). Data protection conformity was confirmed. After consultation with the managing director of the local Institutional Review Board, no vote was required to conduct this study.

In the actual questionnaire, an introductory and explanatory text was followed by the collection of sociodemographic data: Information on gender, age, position within the department or possible retirement and the level of care of the hospital were recorded. The “level of care” is clearly defined for each hospital and set by the respective authorities in Germany. Also the presence of the additional sub-specialty board certifications, such as emergency medicine, pain medicine, palliative care, or intensive care, as well as information on active participation in student training had been queried.

This was followed by the evaluation of a total of 65 practical learning objectives. The order of the practical skills surveyed was based on the conduct of anesthesia and other fields of anesthesiologists’ work. Therefore, the categories were in the order “premedication visit”, “preparation of general anesthesia”, “general anesthesia”, “regional anesthesia and pain therapy”, “intensive care medicine” and finally “emergency medicine”.

A six-point Likert scale (from “not relevant at all” (1) to “very relevant” (6)) was used to determine the relevance each learning objectives at the end of undergraduate training in medical school [18].

### 2.3. Pretest and Survey

A pretest was conducted to validate the questionnaire. For this purpose, 12 participants were interviewed in January 2020. The survey group was mixed in terms of gender and age. Nine anesthesiologists in several hospitals, as well as one non-specialist physician and two non-physician colleagues answered the questionnaire. The pretest was carried out using the same online tool as the actual test and was also evaluated using the same evaluation technique via SPSS (IBM Corporation; New York, NY, USA). The only difference was that a special pretest version of the online questionnaire was used (provided by the SoSci Survey software), which included the possibility for participants to give feedback on each question via extra text fields. We received feedback on technical feasibility, usability, the understanding of individual questions, and the duration of the test. Individual wording was subsequently adjusted and questions supplemented. After this revision, we carried out the online-based survey.

The link to the survey was sent to all members of the DGAI through its official e-mail distribution list. This list contains e-mail addresses of physicians in postgraduate training to become specialists in anesthesiology, specialists and retired physicians (approximately 15,150 e-mail addresses were on file during the survey). The link was not personalized, or password protected. The survey was accessible for a total of three weeks from 11th May to 31st May 2020. The firm deadline was set to increase the response rate as much as possible [19]. One reminder for participation was sent 6 days prior to the end of the survey’s availability by the DGAI.

### 2.4. Statistical Analysis

All data collected during the survey were considered for the analysis. Accordingly, this included the rating of learning objectives of survey participants with questionnaires, which had been filled out completely. All data are presented as median and interquartile range (IQR) as well as mean ± standard deviation (SD) were appropriate. Learning objectives rated with a median of “not relevant at all” (Likert scale: 1) on the 6-point Likert scale were considered irrelevant for the curriculum in the evaluation. In contrast, the learning objectives that were mostly rated as “very relevant” (Likert scale: 6) were considered as essential for future curricula in the field of anesthesiology. To perform analyses of individual groups, normal distribution was determined using a Shapiro-Wilk test. If normal distribution was present, differences between two groups were tested for significance using Student’s *t*-test (or Analysis of Variance (ANOVA) if more than two groups were compared). If the data were not normally distributed, comparisons between groups were performed using a Mann-Whitney-U test (for two groups) or a Kruskal-Wallis test (if more than two groups were compared). Effects were considered significant if p-values were less than 0.05. Statistical analyses were performed using SPSS version 26.0 (IBM Corporation; New York, NY, USA). The figures were created using GraphPad Prism version 9.4.1 for Windows (GraphPad Software, San Diego, CA, USA). The manuscript adheres to the applicable Standards for Quality Improvement Reporting Excellence in Education (SQUIRE-EDU) guidelines [20].

## 3. Results

### 3.1. Data Set and Sociodemographic

During the three-week study period, a total of 2898 questionnaires were returned at least in part, 2046 were completely filled. This corresponds to a response rate of 19%. The number of responses on each day of the survey is shown in Figure 1.

58.94% of respondents were male, 37.51% female, and 0.28% diverse. 3.28% did not provide any information on gender. The age distribution of the participants can be seen in Figure 2.

70.3% (n = 2037) of the survey participants stated that they were holding an additional certification in emergency medicine, 44.1% (n = 1279) were board certified in intensive care medicine. Additional board certifications in pain medicine (13.5%, n = 390) or palliative care (9.8%, n = 285) were less common. 56.8% (n = 1647) of survey participants reported to be actively involved in training medical students. Other demographics, such as personal qualifications or workplace, are shown in Figure 3.

Here, a “hospital with low level of care” is a small hospital with only a few specialties (internal medicine, surgery, anesthesia). A “hospital with advanced level of care” offers a broader range of specialties (e.g., additionally gynecology, pediatrics, neurology, etc.). A “hospital with maximum level of care” offers (almost) all medical specialties independently of a university. A “hospital with special supply mission” is, for example, a specialized heart center.

### 3.2. Overall Rating of Practical Skills

Table 1, Table 2, Table 3 and Table 4 show the individual practical competencies with correspondingly determined median and mean values of the relevance rating.

Six learning objectives were rated as particularly relevant with a median value of 6 on the Likert scale. These six objectives belong to the categories of emergency medicine (“bringing a patient into lateral recumbent position”, “diagnosing a cardiac arrest”, “utilizing the basic life support algorithms”) and preparation of general anesthesia (“setting up an iv-drip for infusion”, “preparing drugs for intravenous application”, “establishing a peripheral iv catheter”). All learning objectives in the regional anesthesia category were rated as irrelevant with a median rating of 1. Furthermore, learning objectives “performing a quick check”, “performing a bronchoscopy”, “performing an minithoracotomy” and “performing a rapid sequence induction” were also considered as not relevant at all.

### 3.3. Rating of Practical Skills Depending on the Different Qualification Levels/Position

Physicians in postgraduate training rated most of the individual practical skills with higher relevance compared to specialists, senior physicians, heads of departments and retired physicians. For example, the learning objective “setting up an iv-drip for infusion” was rated by physicians-in-training with a median of 6, while chief residents reported a median of only 5 here (*p* < 0.001). Furthermore, all regional anesthesia procedures were rated higher by retired physicians compared to all other positions. There was a large difference in the rating among the learning objectives with regard to the subgroup by the skill “transferring information regarding a patient among healthcare professionals utilizing a defined technique (for example SBAR)”: While physicians in training rated it with a median of 5 (mean 4.58/6), all other positions rated it with a median of only 4.

A detailed overview of the rating of the individual learning objectives depending on various factors, such as the individual qualification or the workplace of those questioned, is provided by the tables in the supplement (see Appendix A).

### 3.4. Overview of Significant Differences in Rating between the Different Groups

Colleagues at university hospitals were more likely to rate practical skills as more relevant than physicians working at other levels of care. The vast majority of statistically significant differences were observed between the results of colleagues in university hospitals and physicians providing anesthesia care in outpatient anesthesia practices. Some examples are “performing a 12-channel-ecg and interpretation” with a median of 6 (university hospital) vs. 4 (outpatient anesthesia) (*p* < 0.001), “demonstrating ultrasound examination utilizing the eFAST principle” with a median of 3 vs. 2 (*p* < 0.001) and “evaluating a patient using the ABCDE system” with a median of 6 vs. 5 (*p* < 0.001).

Anesthesiologists, who held an additional qualification in emergency medicine, intensive care medicine, palliative medicine or pain medicine rated the specific knowledge in these topics as less relevant than those colleagues, who did not have a corresponding additional qualification. For example, “taking patient history focused on pain symptoms” was rated less relevant among board certified pain physicians (*p* = 0.025), whereas specialist in intensive care medicine rated “communication adequately with patients and/or relatives in crisis situations” as less relevant than their peers without the additional board certification (*p* = 0.004).

A detailed listing of all significant differences in how the different groups rated the learning objectives is provided in the tables in the Appendix A.

## 4. Discussion

In this study we evaluated an online survey on undergraduate practical skills performed among the members of the German Society of Anesthesiology and Intensive Care Medicine. Some practical skills were considered of utmost importance, such as “utilizing the basic life support algorithms” or “establishing a peripheral iv catheter”. Others were considered to be of less importance. These findings are in accordance with the results of other studies, with a prioritization of theoretical but also of practical skills both in the field of anesthesiology and intensive care medicine [23,24,25]. The overall high relevance of practical skills in this field underscores the importance of our specialty for interdisciplinary student education. Many disciplines are involved in perioperative medical care and could benefit from a better understanding of practical skills [26,27].

The six learning objectives with the highest rating in our study are basic skills like setting up an iv-drip for infusion, i.e. skills that must be mastered by graduates irrespective of specific disciplines [28]. We conclude that skills, which are applicable across disciplines, are of particular importance. The education of medical students should therefore be broad with a prioritization on basic skills [26,29]. The idea of moving away from specialty-oriented training towards pure competency-based training is also reflected in the current reform of medical studies in Germany [30]. The learning objectives assessed as being irrelevant (median of 1), such as “performing a bronchoscopy”, are very specific skills. They play almost no role in everyday clinical practice for physicians of other specialties. The Rapid Sequence Induction is considered a skill that a specialist is required to perform or supervise. This fact is confirmed by the survey results as it was rated as less important for students. All regional anesthesia procedures were also rated as of little relevance. Therefore, these skills should play a subordinate role in the development of a new curriculum. Teaching theoretical basics of these measures seems sufficient and it might not be necessary to perform them independently. In particular, important interdisciplinary practical skills, should be taught by our discipline in the future. The utilization of simulation training might be suitable in this regard, as it could previously be shown that it might increase participants’ self-confidence, efficacy and overall patient safety [31]. We believe that the prioritization of skills in our survey is easily transferable to curricula in other countries. These basic medical skills, which have relevance across many specialties, can be taught well by representatives of our specialty. This is sometimes done at an early stage in medical school so that anesthesiology gains visibility early in the curriculum. The insight that highly specific practical skills from our specialty should not consume time resources in medical school and should rather remain part of postgraduate training certainly applies to the curricula in other countries as well. It is important to use the limited time resources of medical studies for good and repeated teaching and testing of basic key skills rather than superficial training of very specific measures [24].

The consideration of patient safety seems interesting not only with regard to the Helsinki Declaration on patient safety in anesthesiology or the WHO patient safety curriculum guide for medical schools [32,33]. Concern for patient safety is one of the most important skills for anesthesiologists and should already be reflected in medical school curricula [34]. The high rating of concepts such as the SBAR model by anesthesiologists in training shows that the topic is becoming more and more important [35]. This is certainly not only true for Germany. According to a recent meta-analysis, at least one in 20 patients (across all disciplines) are affected by preventable patient harm [36]. Simulation—ideally as a team-based and interprofessional approach—could make a significant contribution to increasing patient safety already during medical studies [37,38]. International comparisons show an increasing role of competence-based learning among universities and their curricula [26,39,40]. This also means that individual specialties such as anesthesiology appear less and less as such in the curricula. Not only in Germany it therefore seems even more important to cover important overarching topics like non-technical skills for patient safety and to establish an association with our specialty already in medical school [41]. In this way, it should be possible in the future to clarify and highlight the professional image and the importance of our specialty even better.

One way to verify that students and physicians are able to perform a particular activity with all its dimensions and facets is via Entrustable Professional Activities (EPAs). EPAs, more specifically, are units of professional practice (tasks or bundles of tasks) that can be fully entrusted to an individual, once they have demonstrated the necessary competence to execute them unsupervised [42]. Especially in recent years, EPAs have increasingly found their way into anesthesiology, so that there are now also publications for anesthesiology EPAs for medical students and novice professionals [43,44]. However, it is important to emphasize at this point that EPAs are not to be synonymous with skills and competencies, but represent a different form of recording and assessment [45].

The evaluation of individual practical skills with higher relevance by physicians in training compared to all other levels of training is striking. A recent study showed that perfectionism, defined as a combination of high standards and high self-criticism, is becoming increasingly common among current medical students [46]. This may therefore also be an explanation for the divergent rating by the physicians in training. Colleagues with higher professional degrees have gained more clinical experience and, as a result, the assessment of skills practiced often and therefore necessary in everyday work is different. A better overview of the entire field as a whole makes detailed knowledge on very specific points seem less important. Furthermore, young graduates may still have problems performing individual competencies themselves resulting in the desire to acquire these prior to the beginning of their specialty training.

Anesthesiologists at university hospitals rated individual practical skills with higher relevance than physicians working at other hospitals. This is also true when comparing university hospital staff to those of outpatient anesthesia practices in the majority of skills. One reason for this could be that, on average, patient comorbidity is higher at university hospitals. This means that many practices are used more frequently and thus become more relevant. Physicians from university hospitals are more often actively involved in the training of medical students. This leads to regular engagement with the teaching of practical skills and thus possibly assigns them greater importance.

Some aspects might act as limitations in our study. Due to the fact that the survey was conducted online, direct enquiries were not possible. However, it was possible to contact the study initiators via e-mail, which was done only five times in total. Therefore, it might possibly be valid to assume that the survey was easy to understand. Furthermore, the link to the survey was not personalized or password-protected. Possible distortions of the results cannot be ruled out completely. However, in our opinion it seems unrealistic that the survey was filled out more than once or answered by colleagues from other disciplines. The low overall response rate of 19% should be seen as another limitation. However, it corresponds to the experience of other online surveys. Paper-based surveys can maybe achieve significantly better response rates if they are completed on site. This at least reflects the authors’ personal experience with respect to student evaluation of teaching (higher response rate with paper-based implementation at the end of the session on-site vs. online evaluation at home). With regard to medical colleagues, such a type of survey would be possible with the aim of achieving the highest possible response rates, e.g., within the context of conferences–but with the increased risk of a selection bias. This is no longer the case if the survey is sent out by post-the latter would not have been realistic for this survey either. The relatively large size of the questionnaire may have had a negative influence on the survey. Nevertheless, we have tried to keep it as simple and understandable as possible. In addition, we again tried to increase the response rate by sending a reminder by e-mail. According to the literature, a second and possibly a third reminder would have been helpful to increase the number of participants again [47,48]. Unfortunately, this was not possible, as the link to the survey was sent via the official e-mail distribution list of the DGAI and a maximum of one reminder may be sent for each survey in order not to burden the recipients with too many e-mails. For possible further studies, one could consider dividing such questionnaires and distributing them over several surveys, whereby the individual surveys would be even more compact. Similarly, holding a parallel lottery for the participants could have a positive effect on the response rate [19,48].

Furthermore, there is a heterogeneous picture in the distribution of sociodemographic aspects, such as age, individual qualifications and workplaces. In terms of the age of participants in the survey, young physicians in training appear to be rather underrepresented, as perceived, and the group of survey participants over the age of 60 somewhat overrepresented. A possible explanation could be that more experienced colleagues and retired physicians might have more time resources or are more motivated to contribute to the further development of their own specialty. According to the authors, the distribution of the participants’ workplaces is fairly representative, with physicians in the outpatient sector and physicians at university hospitals appearing to be somewhat overrepresented. In the latter group, the higher participation is most likely due to the regular contact with students and thus the higher acceptance of the survey. The high rate of survey participants with an additional certification in emergency medicine seems surprising at first glance. However, it should be noted here that this is not a specialist in emergency medicine (this specialty does not exist in Germany). It is an additional certification that entitles the physicians to work preclinically as emergency physicians. Anesthesiologists make up the majority of emergency physicians working in Germany, so this number should not distort the results of the survey.

## 5. Conclusions

In summary our survey provides a good prioritization of practical skills from the fields of anesthesiology, intensive care, emergency medicine, pain medicine and palliative care by experts in the field. The results could therefore be used as the foundation for the development of new curricula and assessment frameworks in Germany and other countries, in which the teaching of practical skills is an important component. Additionally, these results might be able to assist during the establishment of our specialty as a cross-sectional subject in the field of competency-based medical education. In addition to basic anesthesiology and intensive care medicine, topics such as “perioperative care” and “patient safety” play a major role. This might then lead to a potentially greater attractiveness and appeal of our discipline to medical students, who hopefully will decide to become anesthesiologists themselves. In the future, it will be necessary to clarify which teaching formats are best suited to teach each of the learning objectives prioritized here and how the learning objectives can ultimately also be tested in the best standardized way.

## Figures and Tables

**Figure 1 healthcare-10-02260-f001:**
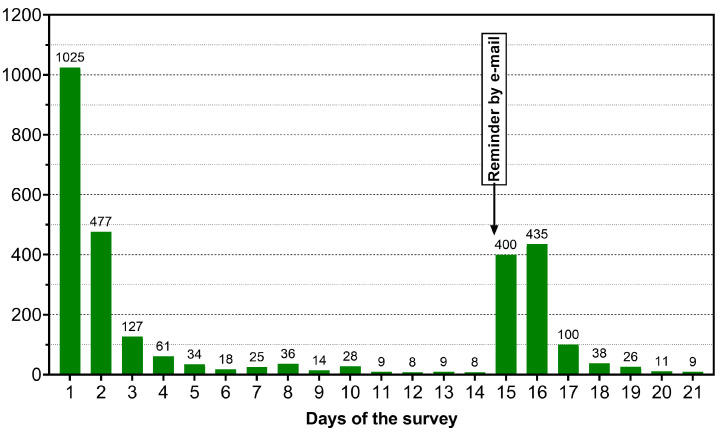
Representation of the number of responses on each of the twenty-one survey days in absolute numbers. The number of responses is plotted above the bars. The time of the reminder six days before the end of the survey was highlighted separately with a text box.

**Figure 2 healthcare-10-02260-f002:**
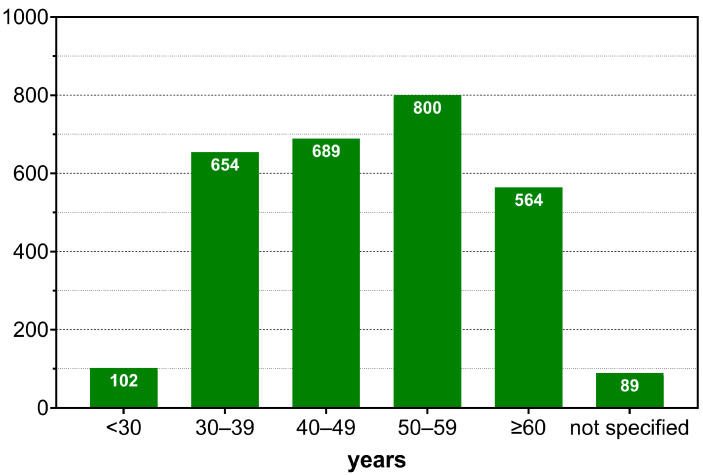
Representation of the distribution of the age of all survey participants (n = 2898) in absolute numbers in the six different categories as a histogram. The number of participants in each age group is plotted at the top of the bars.

**Figure 3 healthcare-10-02260-f003:**
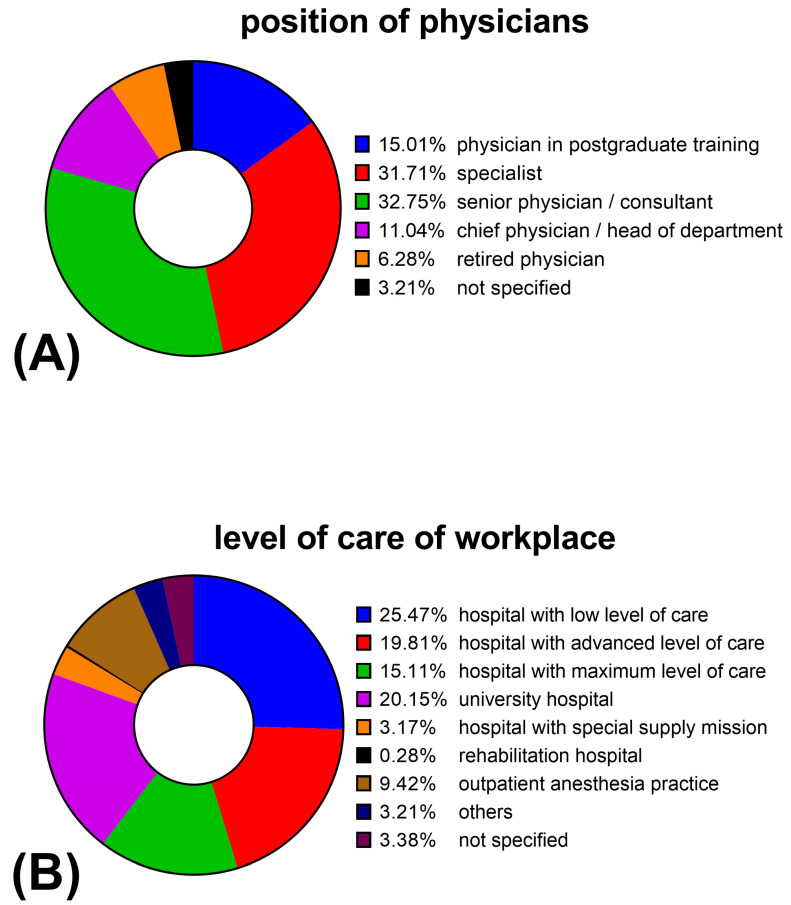
Representation of sociodemographic aspects of the survey participants (n = 2898): Pie chart (**A**) shows distribution of position/qualification of physicians. Pie chart (**B**) shows the physicians’ workplace.

**Table 1 healthcare-10-02260-t001:** Overview of all items asked in the individual categories in the field of anesthesiology (premedication visit und preparation of general anesthesia) and the average rating of the participants from 1 (“not relevant at all”) to 6 (“very relevant”) as median and IQR and mean value and SD.

Item	Category	Competence	Median (IQR)	Mean ± SD
At the end of undergraduate training, the student, as an active member of the professional team, can safely carry out clinical-practical skills adequately and independently under supervision, in a manner that is respectful of the patient. The student can…				
	**premedication visit**			
1	taking patient history relevant to anesthesia		3.00 (2.00)	3.28 ± 1.40
2	performing an anesthesia focused physical examination (auscultation of heart/lung, status of teeth, predictors of a difficult airway, …)		4.00 (2.00)	3.74 ± 1.42
3	performing a 12-channel-ecg and interpretation of the result		5.00 (2.00)	4.62 ± 1.38
4	conducting an informed consent discussion with an ASAI/ASAII patient undergoing a low to medium risk operation and documenting it in a legally correct manner		2.00 (2.00)	2.51 ± 1.35
	**preparation of general anesthesia**			
5	performing a quick check of the anesthesia working place according to the recommendations of DGAI		**1.00** (1.00)	1.95 ± 1.30
6	increasing patient safety by completing a standardized preoperative check list (e.g., WHO check list)		4.00 (3.00)	4.04 ± 1.66
7	establishing intraoperative monitoring (ecg, non-invasive blood pressure monitoring, temperature, relaxometry, pulse oximetry/oxygen saturation)		4.00 (3.00)	4.28 ± 1.49
8	setting up an iv-drip for infusion		**6.00** (2.00)	5.05 ± 1.33
9	preparing drugs for intravenous application		**6.00** (2.00)	4.90 ± 1.41
10	establishing a peripheral iv catheter		**6.00** (2.00)	5.10 ± 1.25
11	establishing a central iv catheter		2.00 (1.00)	1.88 ± 1.07
12	establishing an arterial catheter		2.00 (1.00)	1.85 ± 1.08
13	applying drugs intravenously, intramuscularly, subcutaneously		5.00 (3.00)	4.55 ± 1.52

ASA: American Society of Anesthesiologists Classification; DGAI: German Society of Anesthesiology and Intensive Care Medicine; WHO: World Health Organization.

**Table 2 healthcare-10-02260-t002:** Overview of all items asked in the individual categories in the field of anesthesiology (general anesthesia, regional anesthesia and pain therapy) and the average rating of the participants from 1 (“not relevant at all”) to 6 (“very relevant”) as median and IQR and mean value and SD.

Item	Category	Competence	Median (IQR)	Mean ± SD
At the end of undergraduate training, the student, as an active member of the professional team, can safely carry out clinical-practical skills adequately and independently under supervision, in a manner that is respectful of the patient. The student can…				
	**general anesthesia**			
14	being capable of a sufficient preoxygenation		4.00 (3.00)	3.60 ± 1.64
15	being able to induce a general anesthesia using hypnotics, opioids and muscle relaxants with adequate dosing		2.00 (2.00)	1.96 ± 1.16
16	being able to open the upper respiratory tract by using the Esmarch manoeuvre		5.00 (3.00)	4.51 ± 1.62
17	being capable of ventilating a patient with a face mask (may be using a supraglottic airway tube)		4.00 (3.00)	4.06 ± 1.70
18	knowing how to correctly insert a laryngeal mask airway and checking for its correct positioning		3.00 (2.00)	2.80 ± 1.52
19	knowing how to correctly insert a laryngeal tube and checking for its correct positioning		2.00 (3.00)	2.68 ± 1.57
20	intubating a patient and checking for the correct endotracheal positioning		2.00 (2.00)	2.09 ± 1.23
21	performing the initial steps of an emergency algorithm when encountering an unexpected difficult airway		2.00 (2.00)	2.43 ± 1.50
22	setting up an adequate mechanical ventilation according to the patient and the operation		2.00 (1.00)	1.89 ± 1.12
	**regional anesthesia and pain therapy**			
23	taking patient history focused on pain symptoms		4.00 (2.00)	3.76 ± 1.49
24	setting up a therapy plan according to the WHO analgesic ladder		4.00 (3.00)	3.63 ± 1.50
25	being accustomed to the usage of patient-controlled anesthesia devices		2.00 (2.00)	1.96 ± 1.15
26	performing spinal anesthesia		**1.00** (0.00)	1.33 ± 0.74
27	performing epidural anesthesia		**1.00** (0.00)	1.21 ± 0.61
28	performing combined spinal-epidural anesthesia		**1.00** (0.00)	1.15 ± 0.53
	accomplishing a peripheral nerve block by…			
29	an interscalene approach to the brachial plexus		**1.00** (0.00)	1.14 ± 0.51
30	a supraclavicular approach to the brachial plexus		**1.00** (0.00)	1.13 ± 0.48
31	an axillary approach to the brachial plexus		**1.00** (0.00)	1.19 ± 0.61
32	blocking the femoral nerve		**1.00** (0.00)	1.21 ± 0.62
33	blocking the sciatic nerve with a proximal approach		**1.00** (0.00)	1.14 ± 0.51
34	blocking the sciatic nerve with distal approach		**1.00** (0.00)	1.16 ± 0.53

WHO: World Health Organization.

**Table 3 healthcare-10-02260-t003:** Overview of all items asked in the individual categories in the field of intensive care medicine and the average rating of the participants from 1 (“not relevant at all”) to 6 (“very relevant”) as median and IQR and mean value and SD.

Item	Category	Competence	Median (IQR)	Mean ± SD
At the end of undergraduate training, the student, as an active member of the professional team, can safely carry out clinical-practical skills adequately and independently under supervision, in a manner that is respectful of the patient. The student can…				
	**intensive care medicine**			
35	performing a clinically focused physical exam		5.00 (2.00)	4.71 ± 1.41
36	transferring information regarding a patient among healthcare professionals utilizing a defined technique (for example SBAR)		4.00 (3.00)	4.13 ± 1.61
37	managing an analgosedation for an intervention		2.00 (2.00)	2.15 ± 1.21
38	assessing the depth of sedation of a patient using an established scoring system		2.00 (3.00)	2.68 ± 1.48
39	demonstrating ultrasound examination utilizing the eFAST principle		2.00 (2.00)	2.51 ± 1.50
40	inserting a gastric tube		3.00 (3.00)	3.49 ± 1.61
41	inserting a urinary catheter		3.00 (3.00)	3.47 ± 1.69
42	performing a bronchoscopy on an intubated patient		**1.00** (1.00)	1.40 ± 0.81
43	evacuating air by puncturing of a tension pneumothorax		2.00 (3.00)	2.80 ± 1.72
44	puncturing and/or drainage of intrapleural fluids		2.00 (2.00)	2.03 ± 1.21
45	performing an minithoracotomy and placing a chest tube		**1.00** (1.00)	1.70 ± 1.12
46	obtaining blood samples for microbiological examination		5.00 (3.00)	4.51 ± 1.56
47	performing a blood transfusion according to current guidelines		5.00 (3.00)	4.31 ± 1.77
48	calling a patient’s death		5.00 (3.00)	4.50 ± 1.72
49	inspecting a corpse externally		4.00 (4.00)	3.96 ± 1.86
50	completing a death certificate and correctly differentiating the cause of death		4.00 (4.00)	3.73 ± 1.88
51	communication adequately with patients and/or relatives in crisis situations		4.00 (3.00)	3.72 ± 1.73

SBAR: Situation, Background, Assessment, Recommendation [21,22]; eFAST: extended Focused Assessment with Sonography for Trauma.

**Table 4 healthcare-10-02260-t004:** Overview of all items asked in the individual categories in the field of emergency medicine and the average rating of the participants from 1 (“not relevant at all”) to 6 (“very relevant”) as median and IQR and mean value and SD.

Item	Category	Competence	Median (IQR)	Mean ± SD
At the end of undergraduate training, the student, as an active member of the professional team, can safely carry out clinical-practical skills adequately and independently under supervision, in a manner that is respectful of the patient. The student can…				
	**emergency medicine**			
52	calculating the Glasgow Coma Scale		5.50 (2.00)	4.85 ± 1.43
53	evaluating a patient using the ABCDE system		5.00 (2.00)	4.73 ± 1.49
54	performing a rapid sequence induction and intubation		**1.00** (1.00)	1.79 ± 1.13
55	establishing an intraosseous needle		2.00 (3.00)	2.62 ± 1.60
56	bringing a patient into lateral recumbent position		**6.00** (0.75)	5.45 ± 1.14
57	stabilizing the cervical vertebrae using a stifneck		5.00 (2.00)	4.72 ± 1.53
58	immobilizing a patient using a vacuum mattress or spineboard		4.00 (3.00)	3.92 ± 1.70
59	placing a pelvic binder		3.00 (3.00)	3.40 ± 1.71
60	placing a tourniquet		4.00 (4.00)	3.88 ± 1.73
61	diagnosing a cardiac arrest		**6.00** (0.00)	5.44 ± 1.16
62	utilizing the basic life support algorithms according to current guidelines and performing effective chest compressions		**6.00** (1.00)	5.39 ± 1.19
	utilizing the advanced life support algorithms according to current guidelines and…			
63	correctly analysing the different rhythms in cardiac arrest		5.00 (2.00)	4.61 ± 1.52
64	correctly perform defibrillation/cardioversion		5.00 (2.00)	4.62 ± 1.57
65	correctly administer drugs		5.00 (3.00)	4.55 ± 1.59

ABCDE: Airway, Breathing, Circulation, Disability and Environment/Exposure

## Data Availability

The data presented in this study are available on reasonable request from the corresponding author.

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
