# Peer review of "Teaching Practical Skills in Anesthesia, Intensive Care, Emergency and Pain Medicine—What Is Really Relevant for Medical Students? Results of a German National Survey of Nearly 3000 Anesthesiologists"

_healthcare, 2022, doi:10.3390/healthcare10112260_

Round 1

Reviewer 1 Report

Thank you for the opportunity to review the manuscript of Mr. Gunther Hempel and colleagues titled "Teaching practical skills in anesthesia, intensive care, emergency and pain medicine – what is really relevant for medical students? - Results of a German national survey of nearly 3,000 anesthesiologists". This national survey is very important and relevant for the education of medical students in different fields of anesthesiology.

The manuscript is well written and clearly structured, but I have some minor questions/concerns with the methods, results and discussion of the authors:

Major concerns:

1. Methods, page 3, line 109: Why was the possibility of survey participation limited to three weeks? That seems to me very short. Why only one reminder was used to remind of the survey participation?

2. An overview of the survey participation per week (in the result section) would be very interesting. Possibly, the participation rate increased after the reminder in the third week.

Minor concern:

1. Please explain the abbreviation “SBAR” at the first use (page 7, table 3 or page 8, line 183). For physicians without teaching expertise in undergraduate medical students, this abbreviation is not familiar. Perhaps this study could be helpful: https://doi.org/10.1186/s40886-018-0073-1

2. Page 10, line 295: It was only one reminder by e-mail.

Author Response

First of all, we would like to thank you for your thorough review of our manuscript. We have now incorporated some changes according to your comments that have significantly improved the manuscript. Please find a point-by-point answer to the suggestions below:

Major concerns:

Methods, page 3, line 109: Why was the possibility of survey participation limited to three weeks? That seems to me very short. Why only one reminder was used to remind of the survey participation?

Thank you very much for this comment. We decided to keep the survey period as compact as possible. On the one hand, this decision was due to the fact that distortions in the answers, which could occur due to external influences, should be avoided as far as possible. On the other hand, a long period with repeated reminders would presumably have produced only a few more participants. The figure related to note 2 also shows this very well. There is some evidence that most respondents answer directly or shortly after the invitation or reminder. Setting a deadline also has a positive influence on the response rate (See also: https://doi.org/10.1002/14651858.mr000008.pub4).

We agree with the reviewer regarding the reminder. Here we would have liked to send another one. However, the sending of emails with the link to the survey was officially (and exclusively) done via the professional society, thus allowing only a maximum of one reminder in order not to burden the recipients of the e-mail distribution list with too many messages and reminders.

In order to acknowledge the reviewer’s suggestion, we have added another paragraph regarding this issue starting at line 136 in section 2.3. In addition, we also addressed this aspect separately again in the discussion starting at line 358.

An overview of the survey participation per week (in the result section) would be very interesting. Possibly, the participation rate increased after the reminder in the third week.

Thank you for this comment. We have added a corresponding figure (see figure 1) in order to provide more information on this important topic. Indeed, the reminder briefly led to a significant increase in the number of responses once again.

Minor concern:

Please explain the abbreviation “SBAR” at the first use (page 7, table 3 or page 8, line 183). For physicians without teaching expertise in undergraduate medical students, this abbreviation is not familiar. Perhaps this study could be helpful: https://doi.org/10.1186/s40886-018-0073-1

Thank you for this hint - we have added a corresponding explanation with a literature reference in the legend of Table 3 (same for the other tables).

Page 10, line 295: It was only one reminder by e-mail.

Thank you very much for this note. We have adjusted the wording accordingly (now line 358)

Reviewer 2 Report

Dear authors,

thank you for reporting about this important topic. However, some questions need to be clarified. First, the NKLM is being rated at the present by almost all german faculties- hereby anaesthesiologists are involved. How could your research add benefit to the process which is already conducted?

I think you should stress out more the relevance of practical skills and then transfer to CBME. Practical skills are not CBME! This suggestion has hazards for our future curriculum developments.Therefore, the introduction should be restructured and filled with more aspects.

What is ne "thats new, I can work with that" moment of your work? You did not consider EPAs which are already published for anaesthesiology and made a symbiosis.

Thank you again for dealing with this important topic.

Author Response

First of all, we would like to thank you for your thorough review of our manuscript. We have now incorporated some changes according to your comments that have significantly improved the manuscript. Please find a point-by-point answer to the suggestions below:

First, the NKLM is being rated at the present by almost all german faculties- hereby anaesthesiologists are involved. How could your research add benefit to the process which is already conducted?

Thank you very much for this valuable question. Indeed, an evaluation of selected learning objectives of the NKLM by medical faculties is currently taking place, with feedback being evaluated by separate expert groups in the coming months.

Our study could be a useful addition for the respective anesthesiology clinics. On the one hand, in the prioritization of the teaching of the learning objectives, but also in the selection of additional learning objectives that are only taught individually at the respective location. After all, the NKLM should only contain a core curriculum (80%), which is supplemented by further learning objectives at the individual faculties. In addition, we believe that the results can be used to position the specialty of anesthesiology in new and interdisciplinary learning objectives in the NKLM at the medical faculties.

I think you should stress out more the relevance of practical skills and then transfer to CBME. Practical skills are not CBME! This suggestion has hazards for our future curriculum developments.Therefore, the introduction should be restructured and filled with more aspects.

Thank you for this important comment, which we fully support. Teaching practical skills can only be a component of CBME. We have added a section in the introduction starting at line 59, hoping to better address these important aspects. We have also added further references. At the same time, we have also tried to ensure that the introduction does not become utterly long compared to the manuscript as a whole.

What is the "thats new, I can work with that" moment of your work? You did not consider EPAs which are already published for anaesthesiology and made a symbiosis.

Thank you for this comment. EPAs are another important form of assessment and teaching of competencies - especially when the question arises whether one is capable of doing something. We have added a section to the discussion on this and have also added relevant literature (on the one hand a publication on EPAs in anesthesiology for medical students and another on the differentiation between EPAs and skills/competencies). See line 311.

Reviewer 3 Report

Summary of article

Dr. Busch et al. investigated the priority of teaching each practice skill to medical students for anesthesiologists. Conducting the cross-sectional survey of over 2,000 anesthesiologists in Germany, the authors determined which part of the practice skills should be prioritized. As a result, the authors found that “bringing a patient into lateral recumbent position”, “diagnosing a cardiac arrest”, “utilizing the basic life support algorithms”, “setting up an iv-drip for infusion”, “preparing drugs for intravenous application”, and “establishing a peripheral iv catheter” were the most important parts of the practice skills for medical students. The authors concluded that this survey provides a good prioritization of practical skills for the development of new curricula and assessment frameworks.

Comments (Invitation on Oct 21, 2022, and comment submission on Oct 26, 2022)

This study addressed an interesting topic of medical education in the field of anesthesia. However, I have some concerns for the publication of this manuscript. Please consider addressing some concerns, as shown below.

Here are my comments and suggestions about this manuscript.

Major points:

[1] “Introduction”

Please clarify the motivation for comparing the prioritized practices between the different qualification levels/positions or different groups in the Introduction.

[2] “Method”

Please clarify how the authors define the hospital with a low level of care, advanced level of care, and maximum level of care.

[3] “Result”

In section 3.2, the authors described each practice skill as relevant or irrelevant. This sounds subjective. Therefore, the authors need to define the “relevant/irrelevant” in the Method section.

[4] “Methods”

According to the authors, 2989 participants responded, but only 2046 completed the survey. Please clarify how the author treated the missing data.

Minor points:

[5] “Discussion”

Please add the reference in the sentence, “Paper-based surveys can achieve significantly better response rates (if they are completed on-site).”

Author Response

First of all, we would like to thank you for your thorough review of our manuscript. We have now incorporated some changes according to your comments that have significantly improved the manuscript. Please find a point-by-point answer to the suggestions below:

Major points:

[1] “Introduction”

Please clarify the motivation for comparing the prioritized practices between the different qualification levels/positions or different groups in the Introduction.

Thank you very much for this comment. We have added a section to the introduction starting at line 79. Our aim was indeed to find out whether the prioritization of learning objectives depends on certain factors (age, workplace, etc.) in order to be able to take this into account in the future development of curricula.

[2] “Method”

Please clarify how the authors define the hospital with a low level of care, advanced level of care, and maximum level of care.

Thank you very much for this important comment. The "level of care of the hospital" is clearly defined for each hospital and set by the respective authorities in Germany. Thats why there is no room for individual interpretation. We have made a respective addition in the manuscript from line 105 onwards.

[3] “Result”

In section 3.2, the authors described each practice skill as relevant or irrelevant. This sounds subjective. Therefore, the authors need to define the “relevant/irrelevant” in the Method section.

Thank you very much for this remark. We have made a corresponding addition from line 143 onwards.

[4] “Methods”

According to the authors, 2989 participants responded, but only 2046 completed the survey. Please clarify how the author treated the missing data.

Thank you very much for this inquiry. We have made a corresponding addition to the methodology (see line 140). All data collected were included in the analysis - including questionnaires that were not completely filled out.

Minor points:

[5] “Discussion”

Please add the reference in the sentence, “Paper-based surveys can achieve significantly better response rates (if they are completed on-site).”

Thank you very much for this comment. The sentence represented a personal opinion or experience. It has therefore been reworded accordingly. See line 348.

Reviewer 4 Report

The paper discusses an interesting and potentially useful study. The research methodology is appropriate and relevant and it is good that the authors have validated the questionnaire they used. The paper is in good standing generally, but there are a few areas to consider:

1.      Authors should better justify the study. To do so effectively, they need to present more thoroughly the current situation regarding teaching anesthesiology in Germany, but also elsewhere, and whether practical skills have been incorporated in the curriculum. Has it been incorporated in other countries, eg the UK, and how effective has it been? If there is evidence from other countries, why is it important to know about Germany?

2.      It is good that the authors validated their questionnaire. However, they need to show more statistical evidence as how this was achieved.

3.      Usually, medical curricula are not informed by one study. Which other areas of research has this study opened, which could fill in gaps and well inform medical curricula in Germany and elsewhere?

Author Response

First of all, we would like to thank you for your thorough review of our manuscript. We have now incorporated some changes according to your comments that have significantly improved the manuscript. Please find a point-by-point answer to the suggestions below:

Authors should better justify the study. To do so effectively, they need to present more thoroughly the current situation regarding teaching anesthesiology in Germany, but also elsewhere, and whether practical skills have been incorporated in the curriculum. Has it been incorporated in other countries, eg the UK, and how effective has it been? If there is evidence from other countries, why is it important to know about Germany?

Thank you for this helpful comment. Starting at line 48, we have added a few aspects to the situation in Germany and hope to be able to better highlight the need for and added value of the study. In view of the feedback from the other reviewers, we have also made sure that the introduction is not too extensive and still covers all the required aspects.

It is good that the authors validated their questionnaire. However, they need to show more statistical evidence as how this was achieved.

Thank you very much for this comment. As described in the manuscript, the pretest basically served to check the comprehensibility of the questionnaire - as well as the technical implementation and analysis of the data. The procedure for this is described in the first section of chapter 2.3. A separate statistical evaluation of this validation did not take place. If desired, we would be happy to add the feedback received in detail to the supplement as an additional file. Since the supplement currently already contains 2 files, we would otherwise rather refrain from doing so.

Usually, medical curricula are not informed by one study. Which other areas of research has this study opened, which could fill in gaps and well inform medical curricula in Germany and elsewhere?

Thank you very much for this tip. That is of course completely correct. The results of this study are not yet representative of a curriculum. However, in our view, they do provide an opportunity to allocate and prioritize the available time resources for teaching practical skills in a meaningful way. Other questions that will arise in the future during curriculum development are, for example, with which teaching forms and by whom one wants to convey the respective learning objectives. Likewise, how these learning objectives should best be assessed later in a standardized manner. We have added a corresponding look into the future at the end of the conclusion.

Round 2

Reviewer 2 Report

Thank you for your revision.

Reviewer 4 Report

Thank you for considering my feedback.